# Dose Distribution of High Dose-Rate and Low Dose-Rate Prostate Brachytherapy at Different Intervals—Impact of a Hydrogel Spacer and Prostate Volume

**DOI:** 10.3390/cancers15051396

**Published:** 2023-02-22

**Authors:** Hathal Haddad, Horst Hermani, Herbert Hanitzsch, Albert Heidrich, Michael Pinkawa

**Affiliations:** 1Department of Radiation Oncology, MediClin Robert Janker Klinik, 53129 Bonn, Germany; 2Department of Urology, MediClin Robert Janker Klinik, 53129 Bonn, Germany; 3Urologic Centre Bonn, 53111 Bonn, Germany; 4Curos Urologic Centre, 50389 Wesseling, Germany

**Keywords:** prostate cancer, interstitial brachytherapy, ^125^I, ^192^Ir, dosimetry, spacer

## Abstract

**Simple Summary:**

Two different prostate brachytherapy methods are applied to treat prostate cancer: either a permanent implant of low-dose-rate (LDR-BT) sources or a temporary implant of high-dose-rate (HDR-BT) sources. This study aimed to compare the dose distributions, specifically focusing on the impact of a prostate-rectum spacer and prostate volume. The prostate dose coverage was comparable between different techniques. HDR-BT was characterized by a considerably more homogenous dose distribution and lower doses to the urethra. The minimum dose of up to 5 mm from the prostate was higher for larger prostates. As a consequence of the hydrogel spacer in HDR-BT patients, the intraoperative rectum dose was considerably lower, especially in smaller prostates. However, prostate volume dose coverage was not improved. The dosimetric results explain the clinical differences between the techniques reported in the literature.

**Abstract:**

The study aimed to compare the dose distribution in permanent low-dose-rate brachytherapy (LDR-BT) and high-dose-rate brachytherapy (HDR-BT), specifically focusing on the impact of a spacer and prostate volume. The relative dose distribution of 102 LDR-BT patients (prescription dose 145 Gy) at different intervals was compared with the dose distribution of 105 HDR-BT patients (232 HDR-BT fractions with prescription doses of 9 Gy, *n* = 151, or 11.5 Gy, *n* = 81). A hydrogel spacer (10 mL) was only injected before HDR-BT. For the analysis of dose coverage outside the prostate, a 5 mm margin was added to the prostate volume (PV+). Prostate V100 and D90 of HDR-BT and LDR-BT at different intervals were comparable. HDR-BT was characterized by a considerably more homogenous dose distribution and lower doses to the urethra. The minimum dose in 90% of PV+ was higher for larger prostates. As a consequence of the hydrogel spacer in HDR-BT patients, the intraoperative dose at the rectum was considerably lower, especially in smaller prostates. However, prostate volume dose coverage was not improved. The dosimetric results well explain clinical differences between these techniques reported in the literature review, specifically comparable tumor control, higher acute urinary toxicity rates in LDR-BT in comparison to HDR-BT, decreased rectal toxicity after spacer placement, and improved tumor control after HDR-BT in larger prostate volumes.

## 1. Introduction

Brachytherapy (BT) is associated with a high conformity—a high dose to the tumor region and a steep dose drop-off, with consequential optimum protection of organs at risk (OAR) [1,2,3]. BT for prostate cancer was evaluated in many studies, confirming an excellent local control and acceptable side effects [4,5,6]. It can be applied as a permanent interstitial BT (LDR, low dose rate) and a temporary interstitial BT (HDR, high dose rate) and can also be applied alternatively in many centers [7].

BT is a standard treatment method for prostate cancer in national and international guidelines [1], with a considerably reduced integral dose in comparison to external beam radiotherapy (EBRT). Several studies compared the dosimetry between EBRT and BT [8].

Our study included a large group of patients treated in clinical routine in the last years. A hydrogel spacer was only injected before HDR brachytherapy, as this treatment was usually applied as a boost to EBRT, thus resulting in a higher radiobiological dose and a larger risk for rectal toxicity.

The aim was to analyze and compare the dose distribution in (dose to the macroscopic tumor) and outside (dose to potential microscopic spread beyond the capsule) the prostate between LDR brachytherapy at different intervals (intraoperative, day 1, and day 30 dose distribution) and HDR brachytherapy. The impact of a hydrogel spacer and different prostate volumes was evaluated to specify the additional advantage in respect of the dose to the organs at risk (predominantly dose to the rectum) and the dose to the prostate (possibly improved coverage with a larger distance to an organ at risk). The dose distribution outside the prostate, comparison at different LDR brachytherapy planning and post-planning intervals, evaluation of the advantage of applying a hydrogel spacer, and consideration of different prostate volumes have not been considered in previously published literature and thus improve our knowledge in understanding differences between different brachytherapy techniques.

A literature review was performed to explain the clinical consequences that could result from the dosimetric findings in this study.

## 2. Materials and Methods

The intraoperative dose distribution of 102 LDR-BT (125-I sources with a source activity of 17.8 MBq [0.48 mCi]) patients was compared with the dose distribution of 105 HDR-BT (192-Ir source) patients (232 HDR-BT fractions). The injection of 10 mL hydrogel spacer (SpaceOAR^®^ System, Boston Scientific, Marlborough, MA, USA) was performed under transrectal ultrasound (TRUS) guidance after dissecting the space between the prostate and rectum with a saline/lidocaine solution under local anesthesia, as explained in detail in prior publications [9]. A hydrogel spacer (10 mL) was only applied before HDR-BT, as HDR-BT was commonly applied as a boost to EBRT (resulting in a higher radiobiological dose), whereas LDR-BT was always performed as a monotherapy treatment in a single intervention.

The target volume for all patients was the prostate volume and the base of seminal vesicles without additional margins (CTV, clinical target volume = PTV, planning target volume). The extent of seminal vesicles included was based on individual patient factors, in particular the localization of the tumor in magnetic resonance imaging (MRI) and the localization of positive probes.

The prescription dose was 145 Gy for all LDR-BT patients, using 62 ± 10 (mean ± standard deviation) stranded sources and 21 ± 3 application needles. All patients have been treated with brachytherapy as monotherapy. The prescription dose was 9 Gy (151 fractions, treatment as a boost to EBRT, 1.8–2 Gy single fractions up to 50–50.4 Gy total dose) or 11.5 Gy (81 fractions, treatment as monotherapy in three fractions) for HDR-BT patients, using 14 ± 2 application needles. Treatment planning for LDR-BT was performed using VariSeed 9.0, and treatment planning for HDR-BT was performed using Vitesse 3.0 (Varian, Palo Alto, CA, USA).

The comparison was based on dose levels relative to the respective prescription dose. For the analysis of the dose coverage outside the prostate, a 5 mm margin was added to prostate volume (PV+) as a surrogate of potential microscopic tumor extension and thus potential impact on tumor control. Prostate V100 and V150 denote the PV covered by 100% and 150% of the prescription dose, while prostate D90 and urethra D30 denote the maximum dose in 90% of the PV and 30% of the urethral volume and rectum D2 cm^3^ and rectum D0.1 cm^3^ the maximum dose to 2 cm^3^ and 0.1 cm^3^ of the rectal wall. Urethral volume corresponds to the volume of the transurethral catheter inside the prostate, and the rectum volume corresponds to the anterior rectal wall that could be defined in the transrectal ultrasound. For the comparison of different doses, all dose levels are reported as the percentage of the prescription dose, corresponding to the 100% dose level. Prostate D90 was required to receive at least 100% of the dose in treatment planning (example in Figure 1).

Dose homogeneity indices were calculated to characterize and compare dose homogeneity: Dose Homogeneity Index DHI = (V100 − V150)/V100 (ratio of the target volume which receives a dose in the range of 1.0 to 1.5 times of the reference dose to the volume of the target that receives a dose greater than the reference dose); Overdose Volume Index ODI = V200/V100 (ratio of the target volume that receives a dose equal to or greater than 2.0 times of the reference dose to the volume of the target that receives a dose equal to or greater than the reference dose); Dose Nonuniformity Ratio DNR = V150/V100 (ratio of the target volume that receives a dose equal to or greater than 1.5 times of the reference dose to the volume of the target that receives a dose equal to or greater than the reference dose) [10,11].

Patients after LDR-BT received a post-planning CT on day 1 (with a urethral catheter) and day 30 after implantation. Urethral contour on day 30 was based on estimation.

Statistical analysis was performed using IBM SPSS Statistics 28.0 software (IBM, Armonk, NY, USA). An unpaired t-test was applied to compare volumes and dose levels of different techniques, and a paired t-test to compare volumes and dose levels in LDR-BT patients at different intervals. A chi-square test was applied to compare categorical variables. All *p*-values reported are two-sided; *p* < 0.05 is considered significant.

## 3. Results

As demonstrated in Table 1, patients treated with LDR-BT as a monotherapy are low-risk or favorable intermediate-risk patients who tended to be younger than patients treated with HDR-BT. Many HDR-BT patients are patients with considerable risk factors and more aggressive tumors.

The comparison of volume and dose values is shown in Table 2. HDR-BT patients had a slightly larger PV but considerably larger PV+. The only not significantly different value comparing both methods was prostate V100, whereas prostate D90 was slightly higher in LDR-BT (mean difference of 4%). LDR-BT was characterized by a considerably more inhomogeneous dose distribution, characterized by a prostate V150 twice as high compared to HDR-BT. Accordingly, the Dose Homogeneity Index is less than half in LDR-BT compared to HDR-BT. The Overdose Volume Index and Dose Nonuniformity Ratio are about twice as high in LDR-BT compared to HDR-BT. As a consequence of higher doses within the prostate, considerably higher doses to the urethra resulted (mean difference of 32%).

The injection of a hydrogel spacer between the prostate and anterior rectal wall before HDR-BT resulted in considerably lower doses to the rectal wall. A mean rectum 2 cm^3^ reduction of about 30% and a mean reduction of the maximum dose (rectum 0.1 cm^3^) of about 70% could be achieved in comparison to LDR-BT patients.

Comparing day 1 and day 30 post-planning CT, a significantly lower prostate V150 (55 ± 9% on day 1 and 61 ± 10% on day 30) and prostate D90 (98 ± 8% on day 1 and 102 ± 10% on day 30) resulted on day 1, respectively (*p* < 0.01 for all comparisons). Urethra D30 was also significantly lower on day 1 (114 ± 18% on day 1 and 125 ± 21% on day 30; *p* < 0.01). However, the dose to the rectum increased considerably (rectum 2 cm^3^ of 77 ± 18% on day 1 and 88 ± 21% on day 30; *p* < 0.01). Thus, though differences between LDR-BT and HDR-BT were still highly significant, differences respecting dose inhomogeneity and dose to the urethra decreased in post-planning CT evaluations. However, differences respecting the dose to the rectal wall increased.

In Table 3, the impact of prostate volume on dose distribution is demonstrated (LDR-BT: 52 patients < 41 cm^3^; 50 patients ≥ 41 cm^3^; HDR-BT: 115 fractions < 41 cm^3^; 117 fractions ≥ 41 cm^3^). The dose to the prostate was found to be higher in smaller prostates for LDR-BT but not for HDR-BT. PV + 5 mm D90 was identical for smaller prostates comparing LDR-BT vs. HDR-BT. A difference resulted only in larger prostate volumes, with a considerably higher PV + 5 mm D90 in HDR-BT. PV + 5 mm D90 in HDR-BT was higher for larger prostates in comparison to smaller prostates. However, dose inhomogeneity and dose to the urethra were similar in smaller and larger prostates for both BT techniques; rectal doses tended to be higher in larger prostates (differences between LDR-BT and HDR-BT dose distribution independent of prostate volume). Dose indices were independent of the prostate volume.

## 4. Discussion

### 4.1. Dose Distribution

An adequate radiotherapy dose is needed to achieve adequate local tumor control for prostate cancer [12,13]. Brachytherapy, either as monotherapy or a boost to external beam radiotherapy, is a method of dose escalation for prostate cancer. Several studies, including randomized controlled trials, reported superior local tumor control and biochemical control compared to EBRT alone [6,14,15,16,17]. Higher rectal toxicity rates in combined modality treatments are the rationale for using a hydrogel spacer in this patient group.

In a systemic review including 14 clinical trials with 3534 patients who have been treated with HDR-BT as monotherapy, Viani et al. reported excellent results and effectiveness not only for local control after 5 years but also comparable toxicity rates to EBRT. The mean 5-year biochemical recurrence-free survival (bRFS) for low, intermediate, and high-risk patients were 98%, 94%, and 91%, respectively [18].

In the present study, treatment planning was based on the same average relative prostate V100 (92%). The crucial difference between the techniques was a considerably higher prostate V150 in LDR-BT in comparison to HDR-BT, consequently a lower dose homogeneity and a higher overdose volume, also leading to a considerably higher dose to the urethra. The dose to the rectal wall can be substantially reduced after a hydrogel spacer injection.

Technical or anatomical differences might result in dosimetric differences. Though different mean needle numbers and activities have been used in prior comparisons of HDR-BT vs. LDR-BT dose distributions [19,20], the results of our study are well supported. A former study compared the dose distribution in ultrasound-based planning for prostate cancer brachytherapy as monotherapy with single fraction HDR-BT (19 Gy prescription doses, 192-Ir, and a mean needle number of 18—in contrast to 14 needles in this study) in comparison to LDR-BT (145 Gy, I-125, a median seed activity of 0.56 mCi, and a mean source number of 47—in contrast to 0.48 mCi and 62 in this study). As in our study, relative prostate V150 in LDR-BT was about twice as high as prostate V150 in HDR-BT and urethra D30 was found >10% higher [19]. Mean prostate V150 in our LDR-BT patients corresponded to the objective in a recently published consensus statement (≤65%) [2]. In the publication by Solanki et al., the mean prostate V150 was found to be only about 50% higher in LDR-BT (mean 40%) in comparison to HDR-BT (mean 27%). However, the mean prostate V100 in LDR-BT was only 82% vs. 96% in HDR-BT, thus considerably lower in comparison to our study [20].

Anatomical differences respecting prostate volume have been analyzed in our study in a large patient number. Studies comparing the dose to potential microscopic tumor spread and including postimplant dosimetry have not been published before. A hydrogel spacer has not been used in prior comparisons, and increasing the difference in the dose to the rectum between LDR-BT and HDR-BT has been found to be higher in LDR-BT even without a spacer [19,20].

Nevertheless, a hydrogel spacer (HDR-BT) did not result in an improved PTV coverage that could be assumed with a larger distance between the prostate and anterior rectal wall, as reported in a recently published study using stereotactic body radiotherapy [21]. A similar prostate coverage can be explained by the planning process within our center that is focusing on an adequate, but not increased, minimum dose to the prostate. However, a higher dose resulted in larger prostates with a margin of 5 mm, which is comparable to the PTV definition in EBRT that includes the prostate with a margin. In contrast to EBRT [22,23], potential mitigation of prostate or seminal vesicle motion during treatment in patients with a spacer is less relevant in brachytherapy, as needles and sources are implanted during continuous prostate visualization in TRUS.

In spite of the dose distribution differences reported in our study, additional technical and radiobiological differences need to be considered. In contrast to HDR-BT, the dose in LDR-BT is delivered over a period of many weeks. With increasing and decreasing prostate edema, sources are slightly displaced and dose analyses on day 1 and day 30 differ from intraoperative dose planning [24,25,26].

HDR-BT dose is delivered in a short period and is thus more consistent. However, fractionation concepts often differ between centers. A higher prostate V150 in LDR-BT might be associated with improved tumor control or increased urinary toxicity. Though the dose distribution is important to guide our treatments, the most important results for our clinical practice and guidelines are long-term clinical results.

### 4.2. Association of Dosimetric Results with Clinical Results

Several clinical studies analyzing differences between LDR-BT vs. HDR-BT have been published. Burchardt et al. evaluated differences in a PSA bounce between LDR-BT and HDR-BT and found a similar rate (20–30%) but significantly earlier and shorter PSA bounces after HDR-BT in comparison to LDR-BT [27].

Grills et al. evaluated the potential for differing toxicities between LDR-BT and HDR-BT (38 Gy delivered in four fractions) after a median follow up of 35 months [28]. HDR-BT was associated with decreased rates of acute urinary frequency, urgency, and dysuria compared with LDR-BT. Chronic urinary frequency, urgency, and Grade 2 rectal toxicities were also decreased with HDR. The treatment methods maintained the same biochemical control. A phase 2 randomized pilot study compared PSA outcomes, quality of life, and late toxicity between LDR-BT and HDR-BT (single 19 Gy fraction). Quality of life, including IPSS (International Prostate Symptom Score) and EPIC 26 (Expanded Prostate Cancer Index Composite) urinary irritative scores were significantly more favorable for patients treated with HDR-BT over the first 36 months [3,29]. This can well be the consequence of a lower relative dose to the urethra, as reported in this study. However, a significantly larger proportion of patients after LDR-BT reached PSA < 0.04 ng/mL. This can indicate a larger radio-biologically effective dose, possibly also a result of a considerably larger prostate V150.

In a study comparing LDR-BT vs. HDR-BT boost for external beam radiotherapy, acute urinary toxicity (patient-reported using IPSS and EPIC scores, but also physician-assessed Grade 2 or greater genitourinary toxicity) has been found to be significantly lower (again resulting from a lower dose to the urethra) after HDR-BT boost, though the effect size diminished over time [30].

We have found a significantly higher dose around the prostate (PV + 5 mm D90) in larger prostates—a dose that might cover microscopic tumors around the prostate. In a publication by Le et al. [31], the biochemical control 6 years after HDR-BT was found to be significantly higher for patients with prostate volumes ≥ 60 cm^3^ in comparison to volumes < 60 cm^3^ (91% vs. 78%; *p* < 0.01), though dose prescription to the prostate was found to be comparable. The findings in our analysis could explain these clinical results.

Experience with the hydrogel spacer to reduce the dose to the rectal wall and thus reduce toxicity has also increased considerably in recent years, in particular for EBRT. A systemic review showed not only a significant reduction in the dose to the rectal wall—as demonstrated for our patients in this study—but also reduced the late gastrointestinal and genitourinary toxicity rates [32]. Long-term studies showed significantly improved bowel quality of life for patients treated with a hydrogel spacer [33]. Studies evaluating the advantage of a hydrogel spacer for prostate cancer BT in combination with EBRT or BT as monotherapy have also been published recently [34,35,36,37,38]. A spacer can be advised especially for patients with a higher risk of rectal toxicity [39,40].

## 5. Conclusions

In comparison to HDR-BT, LDR-BT was characterized by a more inhomogeneous dose inside the prostate, a higher dose to the urethra, and a lower dose around the prostate in larger prostate volumes. The dose to the rectal wall could be considerably reduced by applying a hydrogel spacer. However, the prostate volume dose coverage, as determined by prostate V100 and prostate D90, has not been improved with a larger distance between the prostate and anterior rectal wall.

These results explain the clinical differences that have been found in other studies: comparable tumor control (with comparable prostate V100 and dose drop-off around the prostate), higher acute urinary toxicity rates (dose to the urethra), and lower PSA nadir (larger V150 volume) in LDR-BT; higher tumor control in larger prostates after HDR-BT (higher dose around the prostate); and lower rectal toxicity resulting from the application of a hydrogel spacer (considerably lower rectal dose).

## Figures and Tables

**Figure 1 cancers-15-01396-f001:**
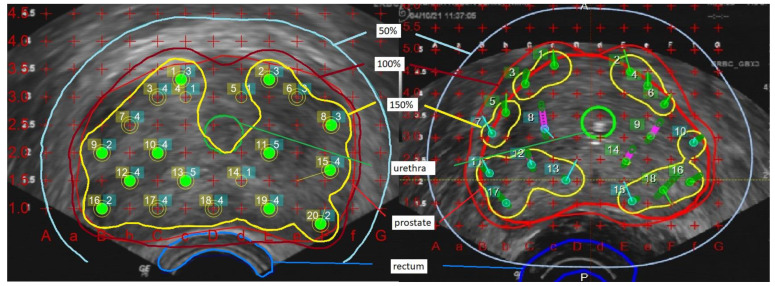
LDR-BT plan and HDR-BT plan with prostate, urethra, rectum contours, 50%-, 100%- and 150%-isodoses (example).

**Table 1 cancers-15-01396-t001:** Patient characteristics.

		LDR-BT (*n* = 102)	HDR-BT (*n* = 105)	*p*-Value
age/years		69 ± 9	72 ± 8	0.06
PSA/ng/ml	<10	91%	61%	<0.01
	10–20	9%	25%	
	>20	0%	14%	
Gleason score	6	54%	10%	<0.01
	7	46%	50%	
	8–10	0%	40%	
T stage	1c–2b	96%	65%	<0.01
	2c–3b	4%	35%	

**Table 2 cancers-15-01396-t002:** Comparison of volume, dose values and homogeneity indices (*p*-value considering comparison of HDR-BT with the respective LDR-BT evaluation).

	HDR-BT (*n* = 232)	LDR-BT Intraop. (*n* = 102)	LDR-BT Day 1 (*n* = 102)	LDR-BT Day 30 (*n* = 102)	*p*-Value Intraop.	*p*-Value Day 1	*p*-Value Day 30
prostate volume	44 ± 17 cm^3^	41 ± 12 cm^3^	40 ± 11 cm^3^	36 ± 10 cm^3^	0.02	<0.01	<0.01
prostate V100	92 ± 5%	92 ± 6%	88 ± 5%	90 ± 5%	0.63	<0.01	<0.01
prostate V150	33 ± 4%	65 ± 9%	55 ± 9%	61 ± 10%	<0.01	<0.01	<0.01
prostate D90	103 ± 6%	107 ± 14%	98 ± 8%	102 ± 10%	<0.01	<0.01	0.60
urethra D30	107 ± 5%	141 ± 20%	114 ± 18%	126 ± 20%	<0.01	<0.01	<0.01
rectum D2 cm^3^	53 ± 9%	69 ± 14%	77 ± 18%	88 ± 20%	<0.01	<0.01	<0.01
rectum D0.1 cm^3^	68 ± 11%	116 ± 32%	138 ± 45%	163 ± 50%	<0.01	<0.01	<0.01
Dose Homogeneity Index	0.64 ± 0.05	0.30 ± 0.08	0.38 ± 0.09	0.33 ± 0.09	<0.01	<0.01	<0.01
Overdose Volume Index	0.15 ± 0.03	0.39 ± 0.08	0.30 ± 0.07	0.38 ± 0.09	<0.01	<0.01	<0.01
Dose Nonuniformity Ratio	0.36 ± 0.05	0.70 ± 0.08	0.62 ± 0.09	0.67 ± 0.09	<0.01	<0.01	<0.01

**Table 3 cancers-15-01396-t003:** Comparison of intraoperative dose values and homogeneity indices depending on prostate volume (smaller or larger than median volume = 41 cm^3^).

		LDR-BT (*n* = 102)	HDR-BT (*n* = 232)	*p*-Value
prostate V100	small	94 ± 5%	93 ± 4%	0.05
	large	90 ± 7%	92 ± 6%	0.05
prostate V150	small	66 ± 9%	33 ± 4%	<0.01
	large	63 ± 7%	32 ± 5%	<0.01
prostate D90	small	111 ± 14%	103 ± 6%	<0.01
	large	102 ± 13%	102 ± 5%	0.84
urethra D30	small	141 ± 20%	107 ± 6%	<0.01
	large	141 ± 20%	108 ± 4%	<0.01
rectum D2 cm^3^	small	68 ± 14%	50 ± 9%	<0.01
	large	71 ± 15%	58 ± 7%	<0.01
rectum D0.1 cm^3^	small	117 ± 36%	65 ± 11%	<0.01
	large	115 ± 28%	72 ± 8%	<0.01
PV + 5 mm D90	small	69 ± 9%	68 ± 6%	0.83
	large	65 ± 8%	73 ± 5%	<0.01
Dose Homogeneity Index	small	0.30 ± 0.08	0.64 ± 0.05	<0.01
	large	0.30 ± 0.08	0.65 ± 0.06	<0.01
Overdose Volume Index	small	0.38 ± 0.08	0.15 ± 0.03	<0.01
	large	0.39 ± 0.08	0.14 ± 0.03	<0.01
Dose Nonuniformity Ratio	small	0.70 ± 0.08	0.36 ± 0.05	<0.01
	large	0.71 ± 0.08	0.35 ± 0.06	<0.01

## Data Availability

The data presented in this study are available on request from the corresponding author. The data are not publicly available, as patient consent has not been given for the publication of individual details.

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
