# Peer review of "Dose Distribution of High Dose-Rate and Low Dose-Rate Prostate Brachytherapy at Different Intervals—Impact of a Hydrogel Spacer and Prostate Volume"

_cancers, 2023, doi:10.3390/cancers15051396_

Round 1

Reviewer 1 Report

Dr. Hathal Haddad and colleagues interestingly described the dose distribution of high and low dose brachytherapy at different intervals and its impact of hydrogel spacer and prostate volume.

More importantly it describes the homogeneous dose distribution inside the prostate. Overall the manuscript written well and scientifically sound.

There are some minor spell checks required throughout the manuscript.

I recommend the manuscript for acceptance after minor spell checks.

Author Response

Thank you for your kind comment. 

The manuscript has been checked for correct spelling.

Reviewer 2 Report

I have read with great interest this article concerning the role of hydrogel spacer for prostate cancer brachytherapy comparing patients treated with or without the use of this device. From a dosimetric point of view, the use of rectal spacer results in a lower dose to urethra and rectum, leading to a more homogeneous dose distribution. Interestingly, there is no PTV coverage improvement. I would suggest the authors to discuss this result, also in light of literature experiences for prostate SBRT in which rectal spacer provides improved target coverage, also for a potential mitigation of prostate motion during treatment. (doi: 10.1259/bjr.20210521; )

No other major observations to notice.

Author Response

Thank you for your kind comment.

In an additional paragraph, the mentioned aspects (PTV coverage, mitigation of prostate motion) have been discussed and references added, as suggested.